# Evolution to alternative levels of stable diversity leaves areas of niche space unexplored

**Ilan N. Rubin**[1]*, **Iaroslav Ispolatov**[2], **Michael Doebeli**[1,3]

**1** Department of Zoology, University of British Columbia, Vancouver, British Columbia, Canada, **2** Universidad de Santiago de Chile (USACH), Departamento de Física, Santiago, Chile, **3** Department of Mathematics, University of British Columbia, Vancouver, British Columbia, Canada

* rubin@zoology.ubc.ca

**Data Availability Statement:** All scripts used to run simulations and generate the data presented in this article are available from https://www.zoology.ubc.ca/~rubin/AltEvoDiversity/.

## Abstract

One of the oldest and most persistent questions in ecology and evolution is whether natural communities tend to evolve toward saturation and maximal diversity. Robert MacArthur's classical theory of niche packing and the theory of adaptive radiations both imply that populations will diversify and fully partition any available niche space. However, the saturation of natural populations is still very much an open area of debate and investigation. Additionally, recent evolutionary theory suggests the existence of alternative evolutionary stable states (ESSs), which implies that some stable communities may not be fully saturated. Using models with classical Lotka-Volterra ecological dynamics and three formulations of evolutionary dynamics (a model using adaptive dynamics, an individual-based model, and a partial differential equation model), we show that following an adaptive radiation, communities can often get stuck in low diversity states when limited by mutations of small phenotypic effect. These low diversity metastable states can also be maintained by limited resources and finite population sizes. When small mutations and finite populations are considered together, it is clear that despite the presence of higher-diversity stable states, natural populations are likely not fully saturating their environment and leaving potential niche space unfilled. Additionally, within-species variation can further reduce community diversity from levels predicted by models that assume species-level homogeneity.

## Author summary

Understanding if and when communities evolve to saturate their local environments is imperative to our understanding of natural populations. Using computer simulations of classical evolutionary models, we study whether adaptive radiations tend to lead toward saturated communities, in which no new species can invade or remain trapped in alternative, lower diversity stable states. We show that with asymmetric competition and small effect mutations, evolutionary Red Queen dynamics can trap communities in low diversity metastable states. Moreover, limited resources not only reduces community population sizes, but also reduces community diversity, denying the formation of saturated

**Funding:** II acknowledges support from FONDECYT (The National Fund for Scientific and Technological Development of Chile) project no. 1200708 (https://www.conicyt.cl/fondecyt/fondecyt-program/). MD acknowledges support from NSERC (The Natural Sciences and Engineering Research Council of Canada) grant no. 219930 (https://www.nserc-crsng.gc.ca/NSERC-CRSNG/Index_eng.asp) The funders had no role in study design, data collection and analysis, decision to publish, or preparation of the manuscript.

**Competing interests:** The authors have declared that no competing interests exist.

communities and stabilizing low diversity, non-stationary evolutionary dynamics. Our results are directly relevant to the longstanding questions important to both ecological empiricists and theoreticians on the species packing and saturation of natural environments. Also, by showing the ease evolution can trap communities in low diversity metastable states, we demonstrate the potential harm in relying solely on ESSs to answer questions of biodiversity.

## Introduction

One of the fundamental goals of ecology is to understand how biodiversity is maintained. Competition theory predicts that two species with identical niches will lead to competitive exclusion and one will win out. Early theoretical work by Robert MacArthur [1–3] codified the idea that competition can lead to the partitioning of continuous phenotype space into niches, allowing for the stable coexistence of species. For species to stably coexist, selective pressures will limit similarity and partition species into individual niches.

MacArthur introduced the idea of niche packing as one of two ways of diversifying (the other being exploration) [1]. Niche packing implies that a higher density of species must lead to a greater partitioning of the available niche space. This has inevitably led ecologists to ask at what point communities will saturate with maximal diversity and whether natural communities tend to exist at saturation [4–6]. While this question has recently led to vigorous debate and research, there has been little theoretical treatment of the evolutionary dynamics for saturated versus unsaturated communities. For instance, it is a well known result that ecological stability does not necessarily imply evolutionary stability and that maximal ecological diversity is often evolutionarily unstable [7].

It has become increasingly clear that eco-evolutionary dynamics play a large role in the long-term maintenance of biodiversity. For example, adaptive radiations, the rapid ecological differentiation of a single clade [8], are able to generate vast amounts of diversity [9, 10]. Eco-evolutionary models of frequency-dependent competition with mutation have been used to show how adaptive radiations can emerge from these simple competitive interactions, leading to diversification and niche partitioning [11, 12].

Recent work has investigated the theoretical existence of alternative evolutionary stable states (ESSs—long-term endpoints of an evolutionary process), the presence of multiple different communities in a given system that are uninvadable by a mutant of small effect [13–15]. The presence of alternative ESSs necessarily implies that certain stable ecological communities may not be at saturation. Certain ESSs may even be "Garden of Edens" that are unreachable by successive small mutations [16] or are only reachable through rapid evolution that occurs on the same timescale as ecology [13, 14]. Calcagno et al. [15] create an atypical scenario running contrary to classical niche-partitioning reasoning, where the initiation of diversification is dependent on there already being diversity present. While both of these results are intriguing, the simple question of whether adaptive radiations tend toward saturated communities or stall at an unsaturated ESS remains largely unanswered.

Given the extraordinary complexity of biological processes, it is natural to think that selection takes place in many dimensions. Despite this, a majority of our intuition of evolutionary dynamics come from narratives of individual traits or models with single phenotypic dynamics. Recent studies of evolution in high-dimensional phenotype space show that evolutionary dynamics can often be complex [17, 18]. With increasing dimension, low and intermediate levels of diversity are increasingly non-stationary, with periodicity most common in lower

dimensions and chaos in high dimensions. As the community diversifies, evolutionary dynamics slow down, often, but not always, fully stabilizing [18]. While the patterns that emerge are often stable, it is not yet clear whether these represent fully saturated communities or lower diversity ESSs. This question is essential for our understanding of how diversity is generated and maintained in natural communities.

Here we investigate whether different patterns of niche partitioning in multi-dimensional phenotype space can lead to alternate levels of stable diversity in a given system and whether adaptive radiations generally lead to saturated communities. We use an eco-evolutionary model with ecological dynamics described by the classic formulation of Lotka-Volterra competition and evolution as a trait substitution process in continuous phenotype space. For computational and visualization simplicity, only two-dimensional phenotypes are considered. The evolutionary dynamics are solved using three separate modeling frameworks: adaptive dynamics [19], individual-based simulations, and partial differential equations. All three models are run with the same ecological dynamics. Numerical simulations of the adaptive dynamics allow for the most efficient computation and most extensive exploration of the three modeling frameworks. The individual-based models allow us to test the assumptions of adaptive dynamics and explore the effects of finite population size and phenotype distributions on the patterns of adaptive radiations.

We will show that adaptive radiations in multiple dimensions often lead to locally stable levels of diversity. Higher diversity states and eventually globally stable community saturation may be reached either through large mutations, immigration of species with phenotypes novel to the community, or if the adaptive radiation was initiated with higher levels of standing genetic variation. With asymmetric competition, the lower diversity states often take the form of stable limit cycles that represent Red Queen dynamics [20]. While these low diversity states are only locally stable, finite population sizes further restrict diversification despite available niche space, stabilizing low diversity, unsaturated communities and, depending on the system, perpetuating Red Queen dynamics. These patterns of locally stable, low levels of diversity are likely present in nature, leaving communities unsaturated and areas of phenotype space open for invasion and continued adaptation. These results shed light on the speed and characteristics of adaptive radiations as well as whether or not diversity tends to evolve to saturation [6].

## Models and methods

### Ecological dynamics

Here we examine a model of phenotypic evolution based on classic logistic Lotka-Volterra ecological dynamics. Individuals are defined by a two-dimensional continuous phenotype and population size. In a monomorphic population of a single phenotype ($\vec{z}$), the equilibrium density of that population equals the carrying capacity, $K(\vec{z})$, of that phenotype. We will use two different forms of carrying capacity functions, from now referenced to as quartic or radially symmetric (Fig 1).

For the quartic case the carrying capacity function is

$$K(\vec{z}) = \exp\left(-\sum_{k=1}^{d} \frac{z_k^4}{4}\right) \tag{1}$$

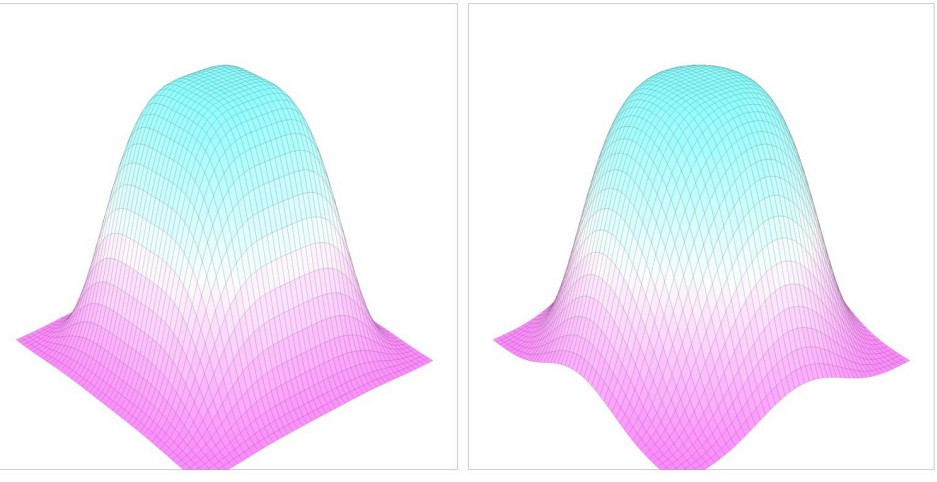

(a)  Quartic carrying capacity

(b)  Radially symmetric carrying capacity

**Fig 1. Carrying capacity functions.** Two carrying capacity functions in 2D trait space. The peak of both functions is 1 at the origin and decreases to 0 as the phenotype increases or decreases. The quartic carrying capacity has a square peak while the radially symmetric carrying capacity has a circular peak. As both functions are of order 4, they are "flatter" on top than a standard Gaussian distribution. For the individual based simulations, the same carrying capacity functions are used, but multiplied by a scalar $K_{max}$ that determines carrying capacity in a number of individuals at the origin. This scalar controls the "richness" of the environment.

The radially symmetric carrying capacity is

$$K(\vec{z}) = \exp\left(-\frac{\left(\sum_{k=1}^{d}\frac{z_k^2}{2}\right)^2}{2}\right) \qquad (2)$$

When evolution is considered, both these functions impose stabilizing selection on the phenotype towards the origin where the carrying capacity is maximal. Due to the higher order term in both of these functions, carrying capacity has a fairly "flat" peak, which naturally restricts viable phenotype space to approximately between −2 and 2 in each dimension. In addition to naturally limiting viable phenotype space, using a quartic carrying capacity avoids the structural instability of a Gaussian carrying capacity and Gaussian competition kernel that results in infinite branching [21, 22].

Individuals compete with others governed by a competition kernel $\alpha(\vec{z}_i, \vec{z}_j)$. The competition kernel equals 1 when an individual competes with another with the same phenotype and decreases to 0 as the phenotypic similarity of the two competing individuals decreases. Thus, similar individuals will have a greater effect on each other's growth compared to individuals with more distinct phenotypes. We consider situations with both symmetric and asymmetric competition. Symmetric competition refers to when individuals impart exactly the same competition load on each other such that $\alpha(\vec{z}_i, \vec{z}_j) = \alpha(\vec{z}_j, \vec{z}_i)$. While this is traditional [21] and conceptually convenient, perfect symmetry rarely occurs in nature and asymmetric competition has been explicitly measured [23]. Therefore, we also consider asymmetric competition where $\alpha(\vec{z}_i, \vec{z}_j) \neq \alpha(\vec{z}_j, \vec{z}_i)$. For symmetric competition, the competition kernel takes the form

of a Gaussian with variance equal to $\sigma_\alpha^2$. For asymmetric competition a term is added with coefficients $b$ that determine the nature of the competitive interaction between the phenotypes in the two dimensions. This asymmetric term is non-mechanistic and can be thought of as the first-order term in a Taylor expansion of some higher order asymmetric interaction function [17]. In this way, it is the simplest way to add asymmetry to Gaussian competition.

$$\alpha(\vec{z}_i, \vec{z}_j) = \exp\left( \sum_{k,l=1}^{d} b_{kl}(z_{ik} - z_{jl})z_{il} - \sum_{k=1}^{d} \frac{(z_{ik} - z_{jk})^2}{2\sigma_\alpha^2} \right) \tag{3}$$

When the $b$ coefficients are set equal to 0, the function reduces to symmetric, Gaussian competition. The competition kernel provides the frequency-dependent component of the ecological dynamics, which allows for the stable coexistence of multiple competing phenotypes under certain conditions.

While the functional forms used here for carrying capacity and competition are largely phenomenological, they represent biologically reasonable scenarios [21] and are well supported in the literature [21, 24].

Based on the classic Lotka-Volterra formulation, ecological dynamics are thus:

$$\frac{dN_i}{dt} = rN_i(t)\left(1 - \frac{\sum_j^M \alpha(\vec{z}_j, \vec{z}_i)N_j(t)}{K(\vec{z}_i)}\right) \tag{4}$$

for growth rate $r$, a population $i$ with phenotype $\vec{z}_i$ and population size $N_i(t)$ that competes with each of $M$ other groups of individuals with distinct phenotypes $\vec{z}_j$, and population sizes $N_j(t)$.

For simplicity and computational tractability, all simulations are spatially homogeneous and positions of individuals or groups in space are ignored.

## Evolutionary dynamics

**Adaptive dynamics.** We model evolution using adaptive dynamics. Adaptive dynamics allows for tractable computation of evolutionary dynamics when a few basic assumptions are met: (1) there is a 1-to-1 map from genotype to phenotype; (2) all genetically identical individuals can be represented as a single phenotype with no phenotypic variation (i.e., a delta function); (3) when a favorable mutation arises, it usually out-competes the resident, driving the resident extinct; and (4) mutations are small and rare. It is essential in the derivation of the adaptive dynamics that the resident populations are at their ecological equilibrium, or in other words, that population dynamics are infinitely fast on the evolutionary timescale and any mutant that arises either fixes in the population or goes extinct before another mutant is introduced.

To derive the adaptive dynamics, we must first define the invasion fitness [25] $f(\vec{z}_r, \vec{z}_m)$ as the per capita birth rate of a rare mutant $m$ in the monomorphic population of resident $r$ that is at its ecological equilibrium population size $K(\vec{z}_r) : f(\vec{z}_r, \vec{z}_m) = 1 - \frac{\alpha(\vec{z}_r, \vec{z}_m)K(\vec{z}_r)}{K(\vec{z}_m)}$. When the invasion fitness is positive ($f(\vec{z}_r, \vec{z}_m) > 0$), the mutant can invade the resident population. When mutual invasibility occurs, i.e., the invasion fitness of the resident into an equilibrium population of the mutant is also positive ($f(\vec{z}_m, \vec{z}_r) > 0$), individuals of the two phenotypes can coexist indefinitely.

By taking the partial derivative of the invasion fitness function with respect to the mutant when the mutant phenotype equals the resident (mutations are infinitely small) we can derive the selection gradient and then the canonical equation of adaptive dynamics, which describes

the trajectory of a single, monomorphic population as it evolves in the trait space (which in our case in two-dimensional). A more detailed description of adaptive dynamics, please refer to Dieckmann and Law [26] and Doebeli [21].

**Speciation.**   The canonical equation of adaptive dynamics, as introduced above (please see S1 Supporting Information for a full derivation of the adaptive dynamics), can thus only describe the movement of populations in phenotype space, but not speciation or extinction events. Evolutionary branching is a well known phenomenon that occurs when there is an attracting equilibrium or nullcline [27–29] in trait space that is also a fitness minimum [19]. To model these we use a well described algorithm [18] where the canonical equation of adaptive dynamics is numerically solved for some period of time at which a random population is chosen and a small mutant is introduced nearby. If that mutant is viable and is mutually invasible with its parent, the population successfully branched and the simulation continues with an additional phenotype. In doing so, this algorithm results in deterministic ecological dynamics and quasi-deterministic evolutionary dynamics in which evolutionary trajectories are deterministic but branching is stochastic.

**Simulations.**   Evolutionary simulations are conducted thusly: (1) solve for the ecological equilibrium of the current population; (2) delete any populations that fall below a minimum viable population size and considered extinct; (3) solve the canonical equation of adaptive dynamics for a fixed length of evolutionary time; (4) for computational speed, merge any two populations whose phenotypes are within a very small distance of each other ($\Delta z$); (5) introduce a mutant population, whose phenotype is a small deviation ($\epsilon_{mut}$) from the parent in a random direction; (6) delete the mutant if it cannot invade the population; and (7) repeat the process until an evolutionary stable community or a designated time has been reached. After each time step, similar phenotypes (those with phenotypes within a small distance from each other) are clustered into species as to better represent the number of distinct phenotypic groups alive at any given time (this does not affect the simulations and is only an accounting device).

## Stability

Given the mathematical complexity of these dynamics, we were unable to analytically determine the evolutionary stability of the resulting communities. Instead, we define metastable evolutionary states, stationary or cyclic, as those, to which the system quickly converges and in which it resides much longer than the convergence time, possibly indefinitely. Mechanistically, the exit from a metastable state is conditional on significantly larger mutations than the convergence to it.

Our definition of metastability is in line with more common causes in physics and chemistry. Consider for example a well-studied problem of protein folding [30]: A model initiated with a completely disordered state quickly folds into one of potentially very many metastable conformations, and it takes long time or a potential guidance (assistance) from other proteins (chaperons) to reach the native functional fold. However, there is a subtle difference between metastability in ordering in physics and metastable states in evolution. In the physics systems, there usually exists a function, such as a free energy, that reaches its absolute minimum in the truly stable state, while the metastable states correspond to local minima, providing a very clear distinction between the former and the latter states. However, in essentially non-stationary systems, such as evolving systems, it is often impossible to define such a function.

## Individual-based model

In order to examine how finite population size affects the evolutionary dynamics, we use an individual-based simulation. In these simulations, individuals have a fixed birth rate and a

frequency-dependent death rate calculated from the carrying capacity and competition kernel as described above. Every time step a single individual is chosen to die or give birth to an offspring with phenotype equal to its parent plus a small mutation in a random phenotypic direction. Instead of a single phenotype describing each population of individuals, populations are now represented by a cloud of points in phenotype space. In order to determine distinct species, individuals are clustered such that every individual of a given "species" lies within a prescribed phenotypic distance to at least one other member of that species. This clustering is purely accounting and has no bearing on the dynamics. For a more detailed account of individual-based simulations and their relation to adaptive dynamics please see Champagnat et al. [31, 32] and Ispolatov et al. [33].

The individual-based simulations are fully stochastic and do not require any of assumptions of adaptive dynamics except the 1-to-1 genotype-phenotype map. Thus, in addition to providing a means to investigate the effects of finite population size, these simulations also act as a check on the applicability of the assumptions required by adaptive dynamics. Additionally, unlike adaptive dynamics, evolutionary branching is an emergent property of the individual-based simulations [31] and acts as a further confirmation of diversification within the adaptive dynamics simulations.

For both adaptive dynamics and individual-based simulations the population is initially seeded with individuals with a randomly chosen phenotype between −2 and 2 in each dimension, roughly coinciding with the area of viable phenotype space for both carrying capacity functions. Simulations were seeded with different numbers of individuals, each with a distinct phenotype, in order to investigate the effects of standing genetic variation on the dynamics of adaptive radiation.

From here on we will refer to groups of phenotypically similar individuals as a species. In the individual-based model a species takes the form of a distinct cloud of points while in the adaptive dynamics simulations a species is represented by a single phenotype and its population size.

## Results

Because the carrying capacity functions we investigate here both have a broader maximum at the origin than that of the Gaussian competition kernel, the origin is always the minimum of the invasion fitness and there necessarily exists a branching point at the origin. Thus the possibility of evolutionary coexistence between two or more phenotypes is guaranteed. In order to further encourage diversification, we only use $\sigma_\alpha < 1$, which would be the condition to generate a branching point at the origin if we had instead used carrying capacity functions of equal order to the competition kernel.

Unless otherwise noted, all figures were generated using parameters listed in S1 Parameters.

### Symmetric competition

For symmetric competition ($b = 0$), a quartic carrying capacity, and $\sigma_\alpha = 0.5$, when starting with a single species (monomorphic population defined by its two dimensional phenotype), that species quickly evolves under directional selection toward the fitness maximum at the origin. Once there, it will start to successively branch, quickly stabilizing into an evolutionary stable community (ESC) with 16 species arrayed in a 4x4 grid (Fig 2). This pattern naturally emerges as the community evolves to pack the viable (and approximately square) niche space, while still maintaining space from neighboring species due to competition. For a more detailed discussion of the patterns and dynamics of species packing in multi-dimensional phenotype spaces please refer to [18].

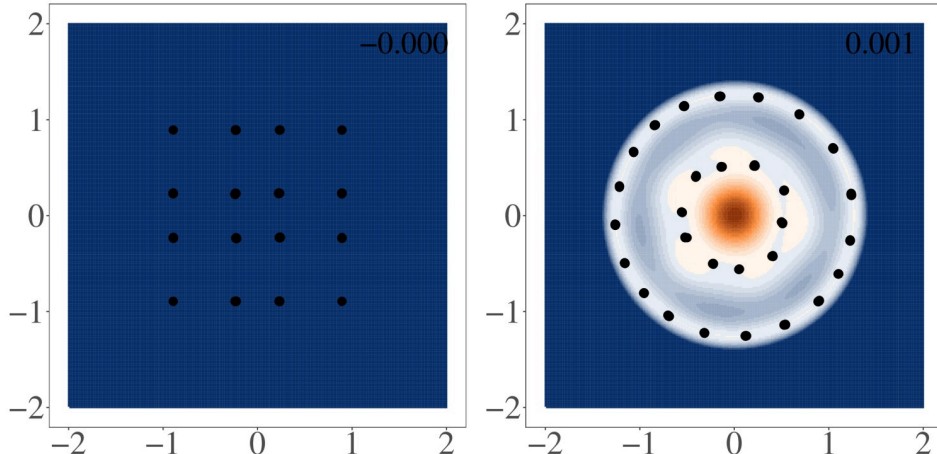

(a) Adaptive dynamics; Quartic carrying capacity

(b) Adaptive dynamics; Radially symmetric carrying capacity

**Fig 2. Stable states for symmetric competition with quartic and radially symmetric carrying capacity.** Figures were generated using adaptive dynamics simulations. Points in the upper panels represent surviving species at the end of each simulation and the surface shows the invasion fitness (per capita growth rate of a rare mutant) with positive invasion fitness displayed in orange and negative in blue. The maximum invasion fitness for each panel is printed in the top right corner. When a symmetric competition kernel was used, simulations all converged on similar patterns regardless of the initial population. All simulations are run with the same parameters, which can be found in Table A in S1 Parameters.

With a radially symmetric carrying capacity, the story is largely the same, but instead of diversifying into a grid, the population arranges itself into two concentric circles. The invasion fitness along the ridges of these concentric circles is almost flat and nearly equal to 0. This resulted in nearly neutral selection along the ridges and different numbers of final species in each simulation, ranging from 24 to 34 between both circles (Fig 2). While the invasion fitness landscape in figure also indicates directional selection toward the origin, the invasion fitness at the origin is three orders of magnitude less than maximum invasion fitness during the initial adaptive radiation, indicating weak selection. Manually placing a population at or near the origin does result in a stable configuration, but diversification toward the origin is so slow it was nearly imperceptible during our simulations even when simulations were run for far longer than our results shown here. Like selection toward the origin, areas of slight positive invasion fitness on the concentric circles indicate that these configurations are likely not fully stable. Clusters on the outer ring seem to be slowly continuing to undergo diversification as long as the simulations were run, though the rate of speciation events and speed of evolution slowed dramatically as the rings filled up.

These simulations with the radially symmetric carrying capacity function and symmetric competition were the only simulations we ran that did not fully settle in stationary or cyclic configurations on timescales that were computationally feasible. Similar long-term transients were not found in any simulations with different functional forms for carrying capacity or competition, or with different parameterizations, so it is likely that this is a degenerate case caused by a perfectly radially symmetric carrying capacity function and Gaussian competition. Indeed, it is a well known result that in a one-dimensional trait space with a Gaussian carrying capacity kernel and symmetric, Gaussian competition, adaptive dynamics results in an infinite branching process [21]. However, like in our simulations, this "infinite branching" quickly

degrades to discrete phenotypes when either the carrying capacity or competition kernels are altered from being perfectly symmetric.

## Asymmetric competition can lead to Red Queen dynamics

For certain values of the four $b$ coefficients (the coefficients that govern the nature of asymmetric competition) the population quickly settles into an evolutionary stable state (ESS). As mentioned in Doebeli et al. [18], in a two-dimensional system like the one simulated here, most randomly chosen $b$ values result in evolutionarily stable communities (ESCs). These configurations are often grid-like for the quartic carrying capacity or concentric circles for the radially symmetric carrying capacity but with some "skew" related to the asymmetry in competition. These ESCs can fully saturate the environment, leaving little to no area of trait space with positive invasion fitness (Fig 2).

Other combinations of $b$ however, can result in non-equilibrium evolutionary dynamics. There is significant literature already detailing how asymmetric competition can lead to non-equilibrium dynamics—either stable limit cycles [34] or in higher dimensions, chaos [17, 18, 35]. These stable limit cycles represent Red Queen dynamics [34] where the community of one or more species continuously evolves (Fig 3). Notably, these cycles are not driven cyclic dynamics in population size, which are not possible with purely competitive interactions and logistic growth, but by the asymmetric competition driving selection around the periodic orbit as can be seen by the selection gradient in Fig 3.

## Red Queen dynamics can lead to alternate levels of metastable diversity

Notably, in simulations that result in stable limit cycles, different numbers of species may emerge (Fig 4). This emergent diversity is then maintained. Unlike an ESC, these stable limit cycles have large areas of positive invasion fitness that are reachable by small mutations–a fact that underlies the non-zero selection gradient and drives the cyclic movement. This also

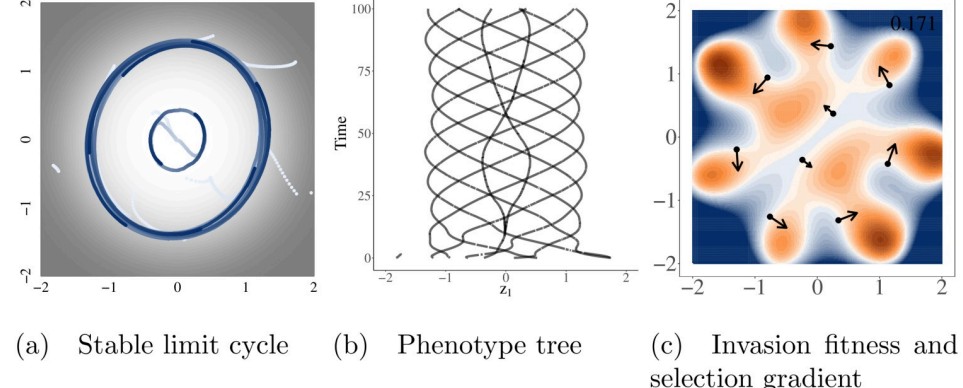

(a)  Stable limit cycle     (b)  Phenotype tree     (c)  Invasion fitness and selection gradient

**Fig 3. Asymmetric competition can lead to Red Queen dynamics.** Red queen dynamics denotes a situation in which one or more populations continuously evolve on a a stable limit cycle in phenotype space. Simulation were run using the radially symmetric carrying capacity and initiated with 10 random species. Panels A and B show the complete history of evolutionary dynamics, with time in panel A increasing from white to blue and carrying capacity increasing from black = 0 to white = 1. Panel C is a depiction of the population at the end of the simulation. Colors in panel C represent the invasion fitness (per capita growth rate of a new mutant if it were to arise). Positive invasion fitness is shown in shades of orange (maximum of 0.17), negative in blue, and invasion fitness equal to zero in white. Arrows are proportional to the square root of the selection gradient for each species. Simulation time was cut to only 100 time steps in the figure so the limit cycles could be more easily seen.

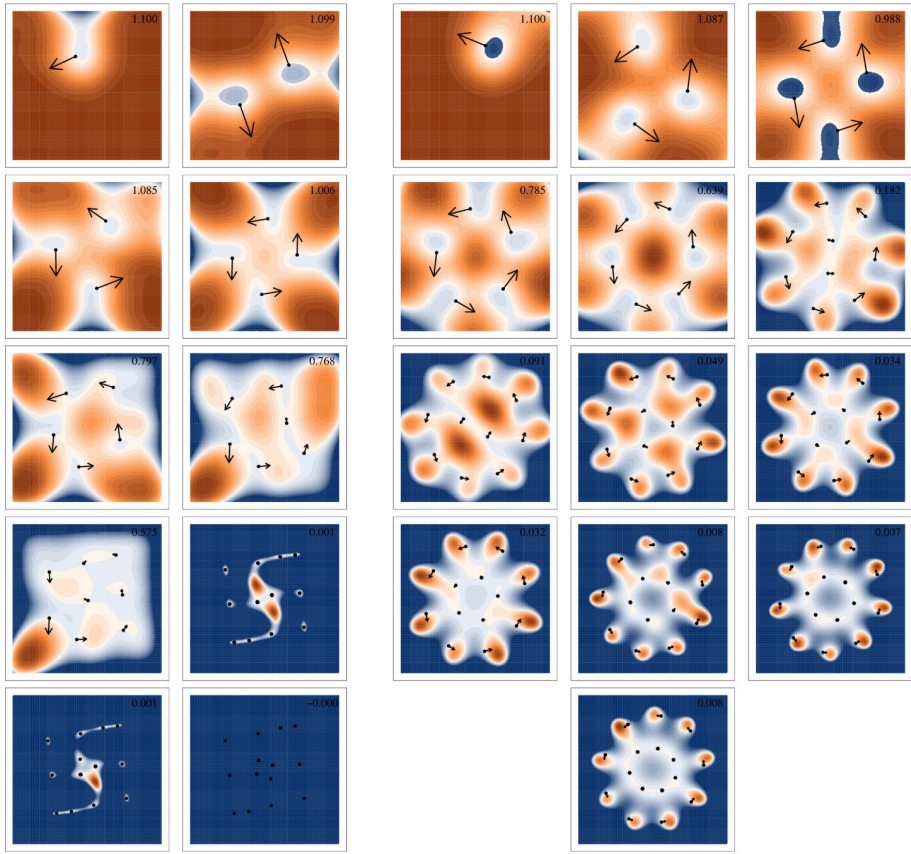

(a)   Quartic carrying capacity  (b)   Radially symmetric carrying capacity

**Fig 4. Invasion fitness landscapes for alternative metastable states.** Alternative metastable states resulting from simulations with different levels of initial population diversity. Each panel represents an alternative level of metastable diversity for a single set of parameters and only differ based on the randomly generated initial communities. Points represent surviving species at the end of each simulation. Arrows are proportional to the square root of the selection gradient for each species. The surface shows the invasion fitness (per capita growth rate of a rare mutant) with positive invasion fitness displayed in orange and negative in blue. The maximum invasion fitness for each panel is displayed in the top right corner of the panel. All axes are displayed from -2 to 2. Simulations in the left two columns use the quartic carrying capacity and those in the right three columns use the radially symmetric carrying capacity. All other parameters can be found in Table A in S1 Parameters. Additional versions of these figures with the initial population also display in addition to the final configuration are linked to in S1 Text.

means that mutants can continuously invade. However, all areas of positive invasion fitness reachable by small mutations only perpetuate the oscillatory motion rather than initiating diversification.

To examine this further, we seeded the adaptive dynamics simulations with an arbitrary number of randomly chosen phenotypes (between 1 and 100 initial species). When seeded with a high number of species, most die out immediately, but many survive. Even after the system is simulated for a long period of time (100,000 branching mutations attempted), those simulations seeded with few species maintain the low diversity while simulations seeded with more tend toward higher diversity stable limit cycles or ESCs (Fig 5). For the parameterization illustrated here and a quartic carrying capacity, metastable limit cycles with 1–6, or 8 species all emerged depending on the initial diversity in the simulation (Figs 4 and 5). Evolutionary

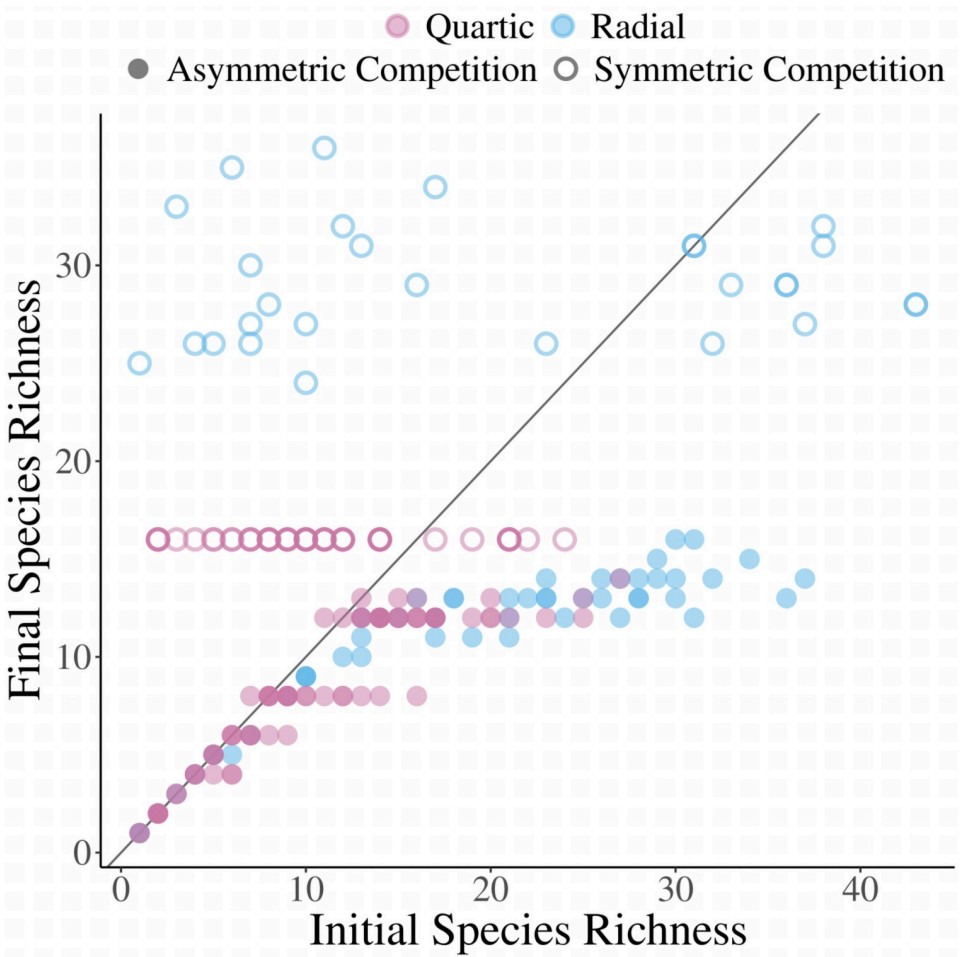

**Fig 5. Levels of metastable diversity.** The final evolutionary diversity when seeding the simulation with different numbers of initial species. Red indicates simulations with a quartic carrying capacity kernel, while blue are those with a radially symmetric carrying capacity kernel. Open circles are simulations with only symmetric competition. Full circles are simulations run with asymmetric competition. The *b* values dictating the competition asymmetry can be found in Table D in S1 Parameters. All other parameters remained the same for all simulations and can be found in Table A S1 Parameters as well.

metastable states of 12, 13, or 14 clusters also emerged when seeded with high initial diversity. Other parameterizations of asymmetric competition (values for *b*) were also tested and while some parameterizations did not result in any cyclic dynamics, alternative levels of metastable diversity were always found. Interestingly, using the quartic carrying capacity, we were unable to find a parameterization of *b* that resulted in a stable limit cycle at low diversity but did not eventually saturate to an ESC if seeded with a diverse initial community.

The presence of a high diversity ESS was not always the case with the radially symmetric carrying capacity. The pattern of low seeded diversity leading to low diversity metastable oscillations is the same, but high diversity stable configurations were often oscillatory as well. Many different simulations with randomly chosen asymmetric competition parameters and the radially symmetric carrying capacity seemingly had no fully saturated ESC, including the parameterization shown here (Fig 4).

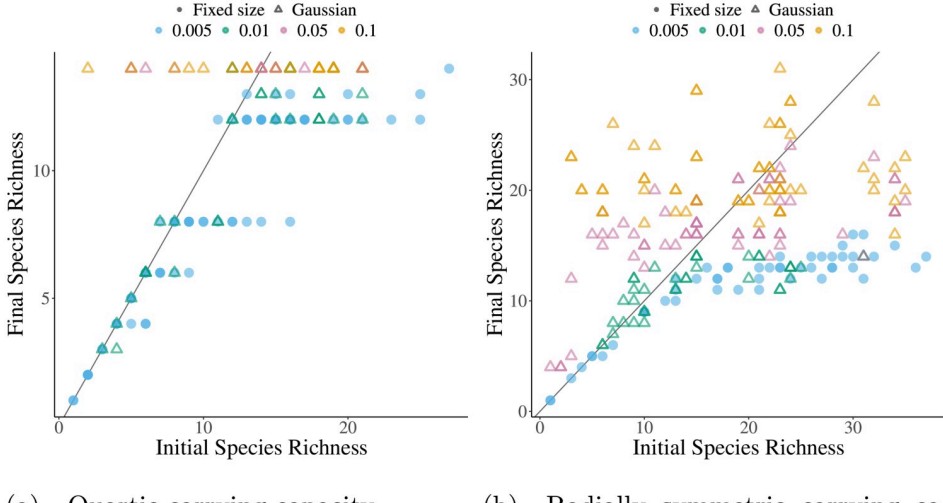

(a)  Quartic carrying capacity          (b)  Radially symmetric carrying capacity

**Fig 6. Large mutations allow escape from low diversity meta-stable states.** The final evolutionary diversity when seeding the simulation with different numbers of initial species for differently sized mutations and asymmetric competition. For solid points, mutants were placed a small fixed distance away from the parents. For hollow triangles, mutations are drawn from a Gaussian with mean equal to the parent's phenotype and standard deviation indicated by point color. Mutation size or standard deviation (depending on the mutation algorithm) are represented by color. All other parameters are the same as those listed in Table A in S1 Parameters.

## Large mutations pushes populations from lower diversity cycles toward saturated ESS

The small mutation assumption of adaptive dynamics is inflexible and necessary in order to derive the evolutionary dynamics. However, branching events are simulated manually. Therefore, we can increase the size of branching mutations (instead of a small fixed mutation $\epsilon_{mut}$, mutant phenotypes are chosen from a Gaussian distribution with mean equal to the parent and standard deviation $\sigma_{mut} \geq \epsilon_{mut}$).

With a quartic carrying capacity, when $\sigma_{mut}$ is increased to 0.05 or larger the lower diversity metastable limit cycles eventually break down and the species-packed ESS is reached (Fig 6). This indicates that these lower diversity cycles are locally stable while the high density ESS is globally stable. For the asymmetric parameters chosen here, the saturated ESS contains 14 species (Fig 4). Of note, because branching mutations are modeled as a Gaussian distribution, even with a small $\sigma_{mut}$ transitions from lower to higher diversity limit cycles did occur, but exceedingly rarely, allowing the lower diversity meta-stable states to persist until the end of our simulations.

For radially symmetric carrying capacity simulations without an ESS, each parameter combination seemingly has a globally stable limit cycle that is eventually reached with large enough mutations. For the parameter values used as an example in this paper, this globally stable limit cycle has 8 species on the outer ring and 5 on the inner, both which cycle clockwise. Unlike with the quartic carrying capacity, there also exist metastable limit cycles with higher diversity than the globally stable one. These "super saturated" communities also collapse to the globally stable cycle when mutation size is increased. Super saturated stable limit cycles occurred rarely when seeded with a random initial species, but were easy to manufacture by manually placing

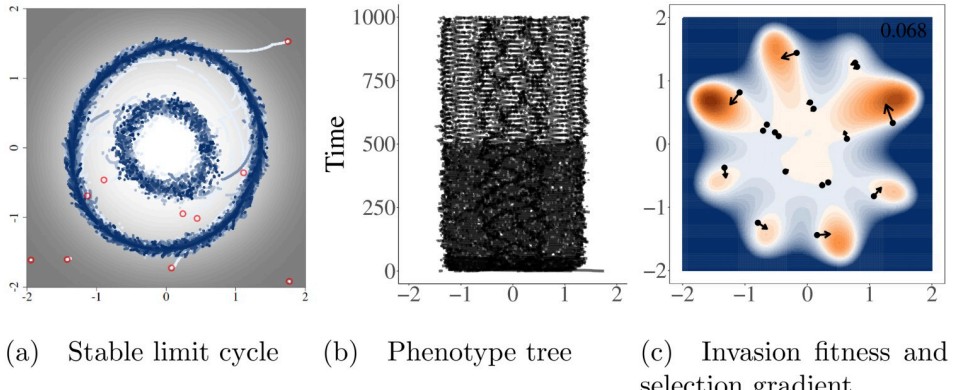

(a)   Stable limit cycle   (b)   Phenotype tree   (c)   Invasion fitness and
                                                        selection gradient

**Fig 7. Large mutations can cause transitions between locally stable levels of diversity.** Simulation were run using the radially symmetric carrying capacity, Gaussian distributed mutations with $\sigma_{mut} = 0.1$, and 10 initial species (randomly chosen). Panels A and B show the complete history of evolutionary dynamics, with time in panel A increasing from white to blue and carrying capacity increasing from black = 0 to white = 1. The initial population is highlighted in red. Transitions between diversity states due to rare, large mutations can be seen in the change in frequency of the limit cycles in Panel B. Panel C is a depiction of the population at the end of the simulation. Colors in panel C represent the invasion fitness. Positive invasion fitness is shown in shades of orange (with a maximum of 0.068), negative in blue, and invasion fitness equal to zero in white. Arrows are proportional to the square root of the selection gradient for each species. Dynamics of the inner circle are under weak selection and provide the environment for mutants to persist for relatively long periods of time.

species in trait space on approximately the two concentric circles that emerge in any community with more than six distinct species (Fig 4).

Because the globally stable community undergoes non-equilibrium dynamics, there still exist areas of positive invasion fitness that drive these cycles (Fig 7). Mutants into these areas of positive invasion fitness eventually out compete the nearby resident, returning the system to 13 species. The invasion fitness landscape around the inner ring is nearly flat, with shallow peaks, leading to nearly neutral local dynamics. This leads to the persistence of small clusters of mutants around the 5 inner species for relatively long periods of time (for a discussion on neutral coexistence see [36]). Despite this apparent increased diversity, the pattern of approximately 13 species clusters is maintained long-term.

## Finite population sizes reduces maximal diversity and facilitates cycling

Individual-based simulations largely aligned with all the results of the adaptive dynamics (Fig 8). Because of the fully stochastic nature of the individual-based model, trajectories obviously didn't match the adaptive dynamics exactly, but the numbers of phenotypic clusters and direction of oscillations in those simulations with limit-cycles qualitatively aligned as expected.

In these simulations, the overall size of the population can be regulated by a parameter $K_{max}$ that controls the height of the peak of the carrying capacity function at the origin. This does not set an artificial cap on the population and instead can be thought of as a parameter controlling the richness of the environment. When $K_{max}$ is large, there are sufficient resources available for a large population to grow. When it is small, resources are exhausted quickly and death rate increases, limiting the size of the population.

When we reduced $K_{max}$, as expected the total size of the population decreased. More interestingly, the maximum number of phenotypic clusters that emerged decreased as well. For example, with a radially symmetric carrying capacity and asymmetric competition, when $K_{max}$ was set to 400, we see 7 or 8 species emerge when starting with high initial diversity (Fig 9).

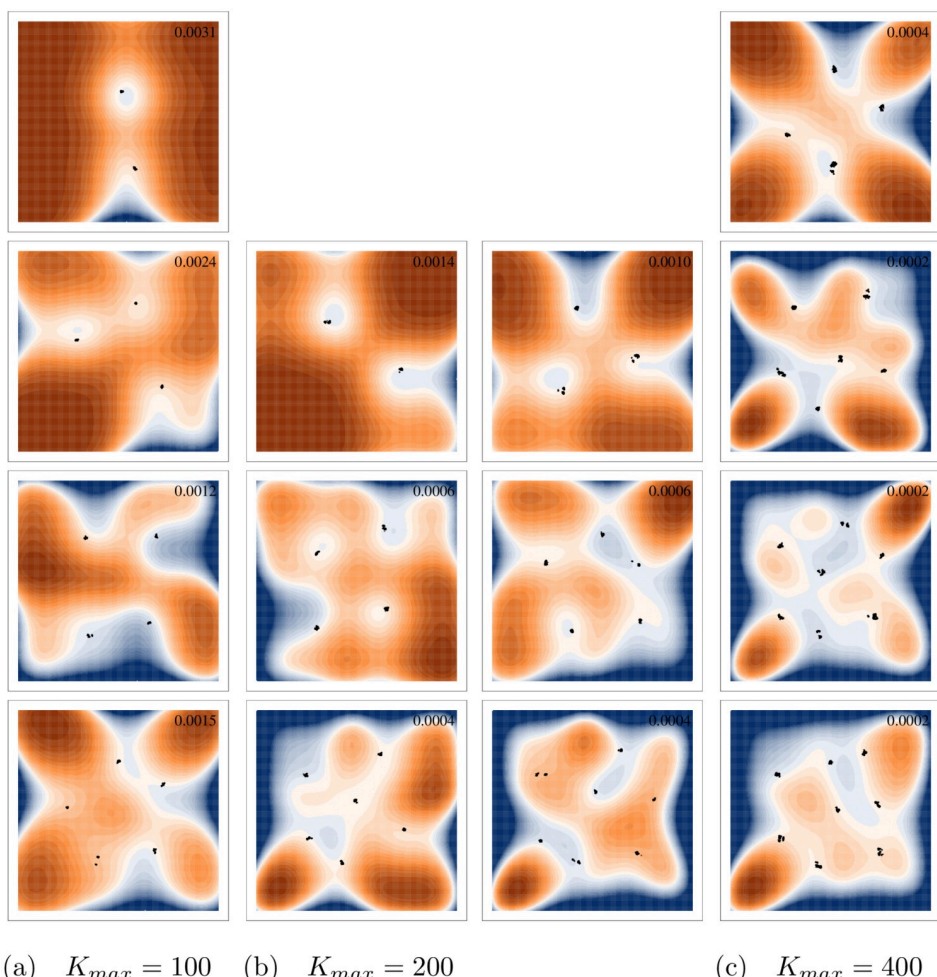

(a)   $K_{max} = 100$   (b)   $K_{max} = 200$                    (c)   $K_{max} = 400$

**Fig 8. Alternative metastable states for different levels of final diversity in individual-based simulations.** Points represent individuals at the end of each simulation. The surface shows the invasion fitness (per capita growth rate of a rare mutant) with positive invasion fitness displayed in orange and negative in blue. The maximum invasion fitness for each panel is displayed in the top right corner of the panel. All axes are displayed from -2 to 2. Simulations use a quartic carrying capacity with $K_{max} = 100$ in the left column, $K_{max} = 200$ in the middle columns, and $K_{max} = 400$ in the right. All other parameters can be found in Table B in S1 Parameters.

However, when $K_{max} = 200$, no more than 6 species were ever maintained for a significant period of time. This was not due to demographic stochasticity, as those 6 species were often maintained in their limit cycle for very long periods of time without extinction or branching. This means that finite resources, and thus finite population sizes, limit the maximal diversity in a system, often facilitating cycling despite the presence of a theoretical global ESS.

Despite the stochasticity, the same general pattern of locally stable, low diversity limit cycles with low initial phenotypic variation and high diversity limit cycles with high initial diversity largely persists (Fig 9). However, with a very low population size (e.g., $K_{max} = 50$, Fig 8), demographic stochasticity overwhelms selection maintaining the low diversity limit cycles, allowing the population to consistently transition to a more stable, higher diversity state. Additionally, unlike in adaptive dynamics simulations, transitions from higher to lower diversity were also possible due to stochastic extinction (e.g., Fig 10).

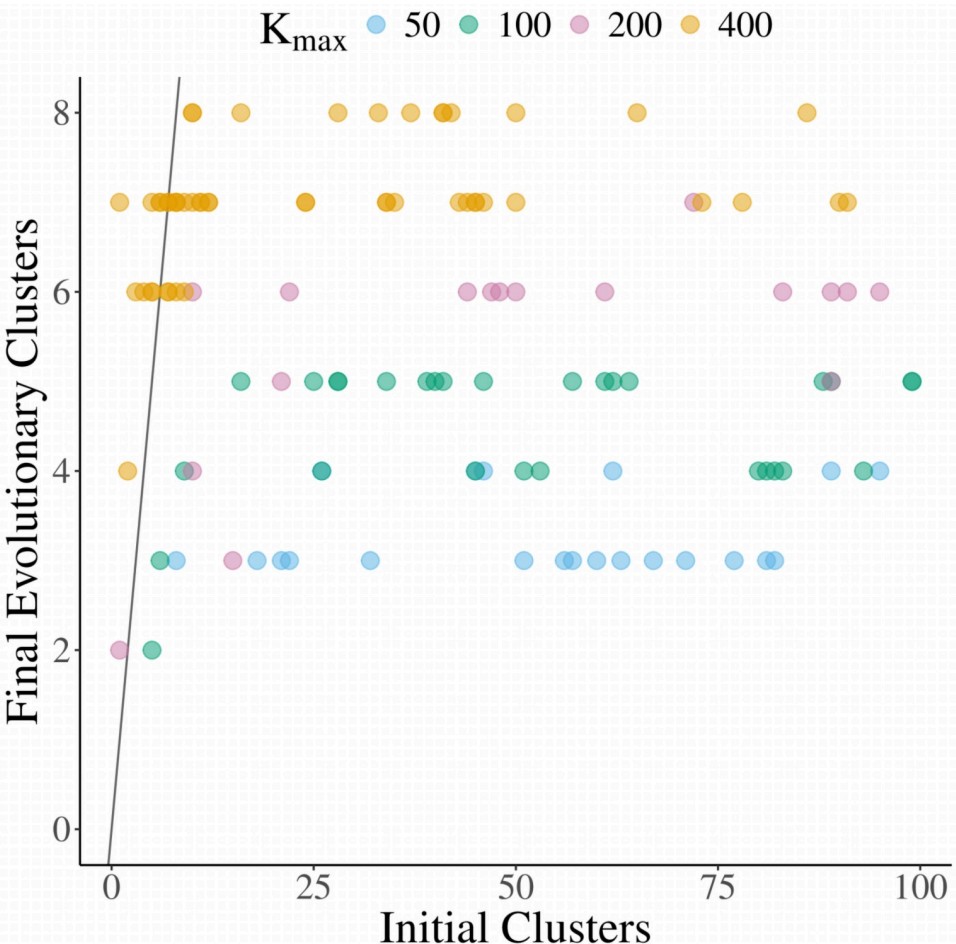

**Fig 9. Finite population reduces realized diversity.** The final number of phenotypic clusters for the individual based model when seeded with different initial population sizes. Individuals were clustered into groups by phenotypic similarity. To control for stochasticity, the final number of clusters was calculated as the median number of clusters over the last 200 time steps. Color indicates the maximum of the carrying capacity kernel (set at the origin). This represents the "richness" of the environment, with larger values modeling an environment with resources that are able to support a larger population. Simulations were run with a quartic carrying capacity kernel and all other parameters remained the same as previous simulations.

## Small population sizes counteracts the effects of large mutations, maintaining low diversity limit cycles

With sufficiently large population sizes ($K_{\max} \gtrsim 400$), increasing the size of mutations moderately had a similar effect to the adaptive dynamics simulations, allowing species to escape lower diversity limit cycles. However, larger mutations also increase the variation within species clusters. When mutation sizes were increased too much, distinct phenotypic clusters all but disappeared with individuals spread out across the entirety of the trait space. Even in these situations the population would cycle in the direction expected if a limit cycle exists.

In the adaptive dynamics, when mutations sizes increased, populations always converged on the most stable configuration, whether that be an ESS or a globally stable limit cycle. With small population size, even with increased mutation sizes, diversity levels remained low. This is because despite there being larger mutations allowing mutants to "jump" across areas of the

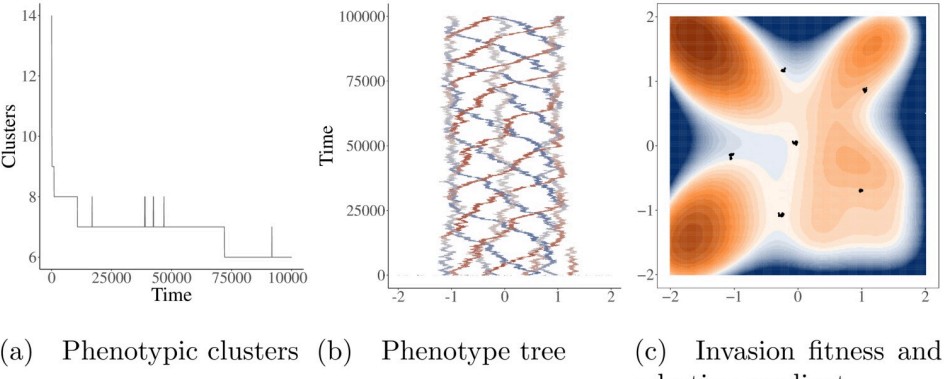

(a)  Phenotypic clusters  (b)  Phenotype tree  (c)  Invasion fitness and selection gradient

**Fig 10. Finite population size can cause transitions in level of diversity due to demographic stochasticity.** This simulation was run using the quartic carrying capacity, Gaussian mutation with $\sigma_{mut}$ = 0.005, $K_{max}$ = 200, and initiated with 89 randomly placed individuals. Panel A displays the number of phenotypic clusters over time. Panel B shows the complete history of evolutionary dynamics, with the x-axis representing phenotype $z_1$ and color representing phenotype $z_2$ (red = -2, white = 0, blue = 2). A transition between diversity states due to a stochastic extinction event can be seen approximately around time = 70000. Panel C is a depiction of the population at the end of the simulation (individuals shown as points). Colors in panel C represent the invasion fitness (measured as the birth rate—death rate of a new mutant). Positive invasion fitness is shown in shades of orange, negative in blue, and invasion fitness equal to zero in white.

negative invasion fitness and diversify, these new mutant species are rarely able to establish. The nature of finite resources inhibiting diversification is strong enough to counteract the ability of large mutations to escape locally stable limit cycles. In these cases, while the number of species was generally maintained, there was increased demographic stochasticity, with populations diversifying and going extinct more often than with smaller mutations.

## Discussion

We have shown here that systems based on Lotka-Volterra competition can cause many different levels of locally stable diversity emerge. This is particularly true with systems of asymmetric competition that lead to periodic evolutionary dynamics in phenotype space. These types of systems often get stuck in locally stable, low diversity limit cycles, despite the presence of a higher diversity global ESS or stable limit cycle.

Classic theory of adaptive radiations expects a quick burst of diversification, followed by a slowdown and possible settling to an ESS—a pattern that has also been shown in natural populations [8, 10, 37, 38]. Adaptive dynamics models show these exact dynamics with successive branching until an ESS is reached [21]. This ESS does not, however, imply community saturation. As expected, our results show that in an adaptive radiation, the population will quickly diversify. If a locally stable ESS or limit cycle is reached, diversification will then come to a stop. However, with small Gaussian mutations, eventually a rare mutation may be introduced that is able to invade the population and another diversification event takes place. This latest branching event could also trigger others, until a new locally stable community is reached. This means that during the early stages of an adaptive radiation, evolution is driven by relatively quick, successive, and small mutations, leading to the expected "early burst" of diversification. However, once a locally stable state is reached, rare, large mutations are necessary to "jump" areas of negative invasion fitness and initiate further diversification, followed by another round of successive small-effect mutations. This large effect mutation mirrors the classic idea of a "key innovation" [39], opening up new areas of adaptive opportunity.

Our findings also compliment findings of non-equilibrium dynamics in higher dimensions [17, 18, 35, 40]. By manually restraining the levels of diversity, Doebeli et al. [18] were able to show that oscillatory and, in higher dimensions, chaotic dynamics are far more likely with lower levels of diversity. Here, we are able to generate the same non-equilibrium dynamics in two dimensions, but as an emergent property of the system. In our results, it became clear that while most systems had locally stable levels of diversity, those with non-equilibrium dynamics were particularly difficult to escape. This is because in non-equilibrium systems, mutations into areas of positive invasion fitness tend to only perpetuate the same cycle. Instead of just having to jump canyons of negative invasion fitness in order to diversify, mutants likely also have to jump across a peak or saddle point of positive invasion fitness. Given the complexity of natural systems, evolution likely takes place in high dimensions. Taken together, these two results imply that many competitive ecosystems are likely unsaturated and undergoing some form of non-equilibrium, or Red Queen, dynamics. Indeed, previous results, both theoretical [41–43] and empirical [44], have implied that Red Queen dynamics may be more generic than previously thought and, like our system, likely stable [45].

Previous theory work on the evolution of diversity via competition for discrete, resources has also supported the notion that evolution often drives ecosystems to remain in an unsaturated state via a common limiting resource [46] or a "diversification-selection balance" with substitutable resources [47]. While these unsaturated states are maintained by different processes than the low diversity metastable states presented here, the diversity of mechanisms promoting low diversity ecosystems hints toward their possible generality in nature.

Perhaps most intriguingly, low diversity states were actually further stabilized by small population sizes. In the individual-based model, reducing the environmental carrying capacity led to smaller population sizes as expected but also a smaller number of phenotypic clusters. For a given value of $K_{max}$ there exists some approximate maximal diversity that can stably exist. This remained true even when mutation size was increased. Increasing diversity with increasing ecosystem productivity has also been confirmed empirically [48]. In the adaptive dynamics simulations, increased mutation size allowed for mutants to jump into areas of positive invasion fitness, causing diversification. In the individual-based simulations this remained true, but these increased levels of diversity would not remain long, with one or more of the species dying out due to small population size stochasticity and increased competition. This means that in small populations, Red Queen dynamics were often perpetuated despite the presence of a higher diversity, globally stable ESS predicted by the adaptive dynamics. These non-equilibrium communities therefore remained in low diversity ESCs for perpetuity.

We should note that increased mutation size can also be considered as a proxy for migration between communities. The lack of a theoretical work on the interplay between eco-evolutionary and metacommunity dynamics is a known problem [49]. Experimental work measuring diversity as a function of immigration history finds that priority effects can play a significant role is shaping resulting communities [48, 50–53]. As the resulting communities are shaped by historical contingencies, this supports the idea that ecosystems may have many locally stable ESSs or limit cycles. Classical metacommunity theory suggests that community similarity will increase with high rates of migration between communities and low total population sizes [54]. Both of these factors proved analogous to our results. Increased mutation size modeling migration allowed communities to escape locally stable ESCs and converge on the unique globally stable ESS or limit cycle. Decreased carrying capacity leading to reduced maximal diversity forces all communities into fewer choices of metastable diversity patterns and therefore increased community similarity.

Niehus et al. [55] propose a model in which horizontal gene transfer acts in accordance with migration to homogenize microbial communities. While we find this idea compelling, we

feel the complexity of HGT and immense diversity in microbial systems necessitate further investigation on the micro-evolutionary dynamics of HGT, representing an important and opportune area for future research.

Ultimately, the model and results presented here are broadly applicable to understanding how frequency-dependent selection can shape the realized diversity of natural communities and the dynamics of adaptive radiations. The pervasive presence of locally stable states or limit cycles in the adaptive dynamics and their further stabilization by finite population sizes suggests at their ubiquity in nature.

## Supporting information

**S1 Supporting Information. Full model details**. Full description of equations, simulation algorithm, and modeling methodology. **Numerical stability analysis**. Numerical stability analysis was performed on each simulation referenced in the text in order to confirm meta-stability. **PDE model**. Evolutionary dynamics when model is formulated as a partial differential equation model.
(PDF)

**S1 Text. Videos, additional figures, and model source code.** Additional supplemental files can be found at https://www.zoology.ubc.ca/~rubin/AltEvoDiversity/.
(PDF)

**S1 Parameters. Table A**. Default parameters for adaptive dynamics simulations. **Table B**. Default parameters for individual-based simulations. **Table C**. Default parameters for partial differential equation based simulations. **Table D**. Default asymmetric competition parameters.
(PDF)

## Author Contributions

**Conceptualization:** Ilan N. Rubin, Michael Doebeli.

**Formal analysis:** Ilan N. Rubin.

**Investigation:** Ilan N. Rubin, Iaroslav Ispolatov.

**Methodology:** Ilan N. Rubin, Iaroslav Ispolatov, Michael Doebeli.

**Software:** Ilan N. Rubin, Iaroslav Ispolatov.

**Supervision:** Michael Doebeli.

**Visualization:** Ilan N. Rubin.

**Writing – original draft:** Ilan N. Rubin.

**Writing – review & editing:** Ilan N. Rubin, Iaroslav Ispolatov, Michael Doebeli.

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
