## [Decision Letter · Decision Letter 0]

8 Mar 2021

Dear Mr. Rubin,

Thank you very much for submitting your manuscript "Evolution to alternative levels of stable diversity leaves areas of niche space unexplored" for consideration at PLOS Computational Biology.

As with all papers reviewed by the journal, your manuscript was reviewed by members of the editorial board and by several independent reviewers. In light of the reviews (below this email), we would like to invite the resubmission of a significantly-revised version that takes into account the reviewers' comments.

We cannot make any decision about publication until we have seen the revised manuscript and your response to the reviewers' comments. Your revised manuscript is also likely to be sent to reviewers for further evaluation.

Sincerely,

Jacopo Grilli

Associate Editor

PLOS Computational Biology

Stefano Allesina

Deputy Editor

PLOS Computational Biology

Reviewer's Responses to Questions

**Comments to the Authors:**

Reviewer #1: In this study, Rubin and colleagues study the process of ecological diversification in a simplified model of ecological and evolutionary competition in a two-dimensional phenotype space. They use this model to ask whether ongoing ecological diversification eventually saturates the available niche space, or whether communities tend to get "trapped" in lower diversity states that are stable against small evolutionary perturbations. By simulating their model under different evolutionary regimes (adaptive dynamics, individual-based simulations, and continuum PDEs) the authors show that some parameter combinations give rise to a range of "metastable" states or limit cycles when initialized with different levels of diversity, while somewhat larger metastable states can be achieved for larger mutation sizes. Based on these observations, the authors conclude that their model communities can often get trapped in lower diversity states, rather than saturating the available niche space.

Overall, I think this is an interesting and important question. My biggest concerns are that the paper felt a bit too descriptive, and that some of the mathematical conclusions -- while plausible -- were not strongly supported by the evidence presented. This negated many of the advantages of authors' highly simplified model, which is otherwise quite far from any real biological system. I think the manuscript would be greatly improved if the authors eliminated some of the more descriptive aspects of the paper and used the reclaimed space to tighten up the argument for the central claim (the existence of alternative stable states below saturation). I describe these and other comments in more detail below.

Detailed comments:

(1) A core concept in this paper is the notion of local evolutionary stability -- i.e., (meta)stability against small evolutionary perturbations. I would distinguish this from the somewhat related concept of stationarity -- the tendency of a system to stay in a given state (or limit cycle) for some long period of time. Metastability and stationarity often go hand in hand, but the latter does not necessarily imply the former, particularly as the dimensionality of the dynamical system increases (as in this study).

My biggest concern is that many of the conclusions in this study are phrased in terms of stability, but much of the evidence is presented in terms of stationarity (e.g. reporting results after some long but finite simulation run). There are some exceptions -- e.g., Fig. 1A convincingly demonstrates stability by plotting the relevant invasion fitnesses. But there are other places (e.g. lines 222-235) where even a demonstrated lack of stability seemed to be equated with a form of "de facto stability". Given the abstract nature of the model, it was not clear why slowness in simulations implies irrelevancy on "reasonable biological timescales" (line 231) without some additional effort to calibrate parameters. E.g. current estimates suggest that a single person's gut microbiota produces anywhere from 1e09 to 1e12 new mutations every day (Zhao & Lieberman et al, Cell Host & Microbe 2019), so ~1e05 attempted branching events does not seem out of the question.

Together, these somewhat fuzzy notions of stability made it much more difficult for me to assess the central claims of the paper, particularly in cases where stable limit cycles started to dominate the dynamics. In these cases, the effective state space of the system (# distinct phenotypic arrangements x # of local mutations) would seem to grow quite large, making it difficult to exhaustively sample the local neighborhood as in Fig. 1A. How can we sure that true instabilities aren't getting swamped by the entropy of the local neighborhood?

My suggestion would be to try to remove some of the more speculative language about stability / the stability landscape (e.g. lines 403 & 417, or the use of "final evolutionary equilibrium" in Figs 4 & 5) or else run some additional numerical experiments that explicitly support these claims.

(2) There is some relevant theoretical literature on the evolution of stable, non-saturating ecosystems that the authors should discuss: Shoresh et al (PNAS, 2008) and Good et al (PNAS, 2019). In particular, the latter study obtained analytical solutions for the evolutionary stable states in a simple resource competition model with large numbers of substitutable resources, demonstrating the existence of a similar Red Queen like state ("diversification-selection balance") that does not saturate the available niche space. Given the apparent similarities with the present findings, the similarities and differences between these studies should be discussed more explicitly in the Introduction and Discussion sections.

(3) In Fig. 3, it would be useful to indicate what the initial population diversity was in each panel, as well as some rationale for how these particular examples were chosen (i.e. at random? to illustrate diversity of behaviors? etc). How are we supposed to interpret apparent similarities (or slight differences) between different panels?

(4) For the cases in Figs. 3 & 4 where the "final evolutionary equilibrium" contains a single strain, would it be possible to analytical verify the stability claim? A few simple examples like this could go a long way toward addressing comment (1) above.

(5) Much of the descriptive feel of the paper (for me at least) arose from the combinatorics of having 2 different carrying capacity models (quartic and radially symmetric) x 2 different types of interactions (symmetric vs asymmetric) x 3 different evolutionary dynamics (adaptive dynamics, individual-based, and PDE). In many cases, it wasn't clear when these models were simply meant to illustrate a diversity of possible behaviors vs proving a more targeted theoretical point.

My suggestion would be to pick one of the scenarios to describe in detail in the main text, and relegate the descriptive aspects of the others to an appendix whenever possible. The reclaimed space could then be used to specifically highlight how these other scenarios support or refute conclusions based on the "main" model alone.

(6) While I agree that the PDE model can be regarded as an infinite population limit of the evolutionary dynamics, a large body of work in simpler systems (e.g. Hallatschek PNAS 2011) has shown that this is often a singular limit in scenarios where populations are continually adapting through new mutations. Thus, it is not necessarily clear that the PDE model should be a good approximation in large but finite populations, as implied by the discussion on 467.

Reviewer #2: In the submitted paper, Rubin et al study if eco-evolutionary systems with adaptive radiations lead to fully saturated communities. In particular, they use different simulation approaches to study the problem and find out that communities can often end up in states where the niche space is not fully saturated.

The article is interesting, and I think that it has the potential to be published. There are however a few (minor) adjustments that should be done before granting publication.

These are my comments, from the most to the least important:

1) While the description of *what* the authors do when building the models is clear, the reasons behind their choices are not always as clear. For example, what is the reason behind the choice of the functional forms used for the competition functions α and for the carrying capacities K in Eqs (2)-(6) (beyond the ease of computation)? In other words, why did the authors choose a Gaussian form for the competition function, and quartic functions for the carrying capacities? Are there (even very simple) practical examples where these functional forms could be applied? I think that the article would gain a lot in interest to potential reader if these aspects were discussed more in depth.

2) Related to the previous point, why is there a particular reason why the phenotype space that the authors consider is limited to [-2,2] in both directions?

3) The authors do a very good job at describing the results that they have found. There are however some parts where I think a more thorough interpretation of the results would be desirable. For example, in the paragraph "Symmetric competition" (from line 214), is there any way to understand why the authors are seeing what they see? For example, is there a way to understand why the final state in phenotype space is a 4x4 grid, and why the points in the grid are in those exact positions? Do these properties depend on the particular values of the parameters? And if so, how? Is there any "universal" (i.e., not depending on the parameter values) property of the system in this case?

4) The paragraphs "Ecological dynamics" and "Evolutionary dynamics" under "Models and methods" (from line 76) are a bit cryptic at first reading, because several aspects of the models are discussed without showing the equations and the formulas to which the text refers to. Furthermore, some parts that are actually useful to understand how the models work are relegated in the Supplementary Information. I therefore suggest the authors to move the description of the properties of the models from the Supplementary Information to this part of the paper. I think that this way the article would gain considerably in clarity, particularly at first reading.

5) I have not understood the reason behind the choice of Eq (12) as the form for the death rate in the individual-based model. I think the authors should explain why they have chosen this form more in detail.

6) I think that the paragraph "Individual-based model" (from line 166) would be more clear if the authors spent a few words explaining how the clustering of individuals is made (even if they cite a few articles on line 178)

7) On lines 77-78, the authors say that they use "classic logistic consumer resource ecological dynamics". This choice of words, and in particular "consumer resource", is potentially misleading since it could refer to MacArthur's consumer-resource framework for modelling population dynamics. I therefore invite the authors to say "classic logistic Lotka-Volterra ecological dynamics".

8) There are some typos throughout the text (e.g., "stats" instead of "states" at the end of the Author Summary, "affect" instead of "effect" in line 92, "α(z_i,z_j)=α(z_i,z_j)" instead of "α(z_i,z_j)=α(z_j,z_i)" in line 96, etc.). I recommend the authors to check again the whole text for typos.

Reviewer #3: I've given the manuscript "Evolution to alternative levels of stable diversity leaves areas of niche space unexplored" by Ilan Rubin and colleagues a careful reading. In it, the authors use three eco-evolutionary frameworks to explore how asymmetric competition along two trait axes can lead to evolutionary limit cycles that prevent the ultimate evolutionarily stable community from being reached. It's an intriguing premise that I was eager to read about. The technical simulations seem carefully done (I was able to recreate the results of theirs that I attempted). All in all, a thought-provoking paper!

That said, there were a number of places where the writing was imprecise, which I will outline below. These issues can probably be easily addressed. I also wonder if the generality of the results is somewhat oversold, but I still think the paper highlights some interesting issues.

Some of the wording in the introduction seems off. E.g.

l.2-3: "how different patterns of biodiversity are maintained" or "how biodiversity is maintained"?

l.6-7: "stable coexistence of diversity" or "stable coexistence of species"?

l.7: "For communities to coexist" or "For species to coexist"

I suggest a careful reworking of these and other phrases throughout.

l.9-10 This paragraph starts with talking about two ways of diversifying, but we never hear about "exploration". Why introduce it?

l.27: when you say "alternative ESS's", do you mean local evolutionary stability or global?

l.52: "trait substitution process" evokes the stochastic process followed by Geritz et al. 1998, not the smooth gradient dynamics used here

l.64: introduce the alternative explanation for large mutations (species invasion) already here

l.98: The phrase "asymmetric competition" is usually taken to mean when a larger trait value is better -- "Asymmetric competition arises when, during an encounter between two or more individuals for some limited resource, these resources are divided up unequally: the larger individual wins the contest, the territory holder keeps the territory, the taller plant gets more light." from ref [31]. That is not how it is used in this paper, which was a major point of confusion for me until I explored the competition function myself. This should be at least pointed out, but preferably a different phrase used.

model description: I think it would help to give the equation of the competition kernel in the main text, and to have a figure with it and the two carrying capacity functions

There seems to be contradiction between l.122 "when a favorable mutation arises, it out-competes the resident" and l.134 talking about potential coexistence of mutant and resident. Maybe add "usually" to l.122?

l.143 "adaptive dynamics" is often used to model branching/"speciation", so why say it can't be?

l.146 "well described algorithm" -- well described where?

l.206 not clear what "higher order" means here, or where it was shown that this implies branching. Got a reference?

l.310 define "metastable"

Fig. 4-5: what does "initial ecological equilibrium" and "final evolutionary equilibrium" on axes mean? initial & final species richness might be better

l.386 this isn't really a resource competition model, more phenomenological

l.419 It's a leap to suggest that most natural systems are unsaturated and undergoing Red Queen dynamics based on this model with MANY unrealistic assumptions (no immigration, Lotka-Volterra competition, etc.)

eqn (8): is that \\partial f or \\partial \\alpha?

l.526 it's well known that the relative evolutionary rates can affect the stability of evolutionary equilibria. Am I missing something?

eqns (10) & (11): how do these follow from (7) and (8)? seems like there are many terms missing

When I attempted to replicate the results (quartic fitness function), I found that periodically invading the co-evolutionary limit cycles could indeed build up to the final ESS. What did I do differently? When I introduced new species, I did not first check that they had positive invasion fitness, but instead just followed their own trait (and population) dynamics. They easily diverged from each other, and never reached low population densities along the way either. After 7 species, the limit cycling was replaced by an evolutionary equilibrium, which ended up with 9 species in a skewed grid. So, perhaps a hidden assumption required to find the authors' metastable cycles is the strict separation of ecological and evolutionary time scales. Worth looking into!

**Have all data underlying the figures and results presented in the manuscript been provided?**

Reviewer #1: None

Reviewer #2: Yes

Reviewer #3: None

PLOS authors have the option to publish the peer review history of their article (what does this mean?). If published, this will include your full peer review and any attached files.

Reviewer #1: No

Reviewer #2: No

Reviewer #3: No
---

## [Decision Letter · Decision Letter 1]

16 Jun 2021

Dear Mr. Rubin,

Thank you very much for submitting your manuscript "Evolution to alternative levels of stable diversity leaves areas of niche space unexplored" for consideration at PLOS Computational Biology. As with all papers reviewed by the journal, your manuscript was reviewed by members of the editorial board and by several independent reviewers. The reviewers appreciated the attention to an important topic. Based on the reviews, we are likely to accept this manuscript for publication, providing that you modify the manuscript according to the review recommendations.

Sincerely,

Jacopo Grilli

Associate Editor

PLOS Computational Biology

Stefano Allesina

Deputy Editor

PLOS Computational Biology

[LINK]

Reviewer's Responses to Questions

**Comments to the Authors:**

Reviewer #1: I'd like to thank the authors for their work in revising the manuscript. Most of my comments have been addressed, except for one remaining point on the PDE model. I agree with the authors that this is not of central importance to the revised manuscript, but for the sake of technical accuracy in the Appendix, I think it is still important to address.

I had a little trouble following the logic in the authors' response, so I'll try to rephrase: I'm not trying to argue that the large N limit is unrealistic or not worth looking at, but rather that as a technical matter, the N=infinity PDE model is *NOT* necessarily the same as the N->infinity limit of the individual-based model. I.e., in an asymptotic sense, the leading order contribution as N->infinity is not necessarily the same as setting N=infinity from the start. This is known to be the case in the simple traveling wave models studied by Hallatschek and others, and I don't see why we'd expect it to be any different for the more complicated models studied here.

With that in mind, my only issue w/ the phrasing in the revised manuscript is line 700: "The large population limit of the individual based model gives the deterministic formulation of partial differential equations." A simple fix would be to change the beginning to "the infinite population version" or "the deterministic limit" instead.

Reviewer #3: Reviewer #3 from the previous round here. This remains an interesting study on the existence of local ESS's and Red Queen evolutionary limit cycles in eco-evolutionary models, which I'll be glad to see in print.

That said, some of my comments from the previous version were not accepted, which I think will lead to some confusion for some readers. Particularly, the authors take "adaptive dynamics" to mean only directional selection of extant species, not the initiation of new lineages through evolutionary branching. This deviates from the commonly accepted usage starting with the foundational papers of adaptive dynamics (Metz et al. 1996, Geritz et al. 1998). It leads to contradictory statements like "when a favorable mutation arises, it out-competes the resident" (l.130) and "when mutual invasibility occurs ... the two phenotypes can coexist indefinitely" (l.140-142) and "unlike adaptive dynamics, evolutionary branching is an emergent property" (l.210). The fix is simple: add "usually" to l. 130 and don't use "adaptive dynamics" as a synonym for "the canonical equation of adaptive dynamics" (e.g. l.155, 164, and throughout).

Other comments:

l.72 The conclusion that "locally stable, low levels of diversity are likely ubiquitous in nature" is a bit strong. I think the authors have show that it is possible, but it's certainly not the only outcome in the authors' models, which also rely on particular assumptions.

l.104 reverse the subscripts in the second part

l.150 The canonical equation isn't described above.

l.234 fix \\alpha

Fig. 2 Put the range of the axes on the axes, not in the figure legend.

l.246 For clarity, I'd say "almost flat and nearly equal to zero"

throughout - suppress unintentional new paragraphs after equations with \\noindent (e.g. l.87, 111, ...)

**Have the authors made all data and (if applicable) computational code underlying the findings in their manuscript fully available?**

Reviewer #1: Yes

Reviewer #3: Yes

PLOS authors have the option to publish the peer review history of their article (what does this mean?). If published, this will include your full peer review and any attached files.

Reviewer #1: No

Reviewer #3: No

Figure Files:

Data Requirements:

Reproducibility:

References:

---

## [Editor Report · Decision Letter 2]

7 Jul 2021

Dear Mr. Rubin,

We are pleased to inform you that your manuscript 'Evolution to alternative levels of stable diversity leaves areas of niche space unexplored' has been provisionally accepted for publication in PLOS Computational Biology.

Best regards,

Jacopo Grilli

Associate Editor

PLOS Computational Biology

Stefano Allesina

Deputy Editor

PLOS Computational Biology

---

## [Editor Report · Acceptance letter]

23 Jul 2021

PCOMPBIOL-D-20-02301R2 

Evolution to alternative levels of stable diversity leaves areas of niche space unexplored

Dear Dr Rubin,

I am pleased to inform you that your manuscript has been formally accepted for publication in PLOS Computational Biology. Your manuscript is now with our production department and you will be notified of the publication date in due course.

With kind regards,

Zsofi Zombor
